# Current Psychological Distress, Post-traumatic Stress, and Radiation Health Anxiety Remain High for Those Who Have Rebuilt Permanent Homes Following the Fukushima Nuclear Disaster

**DOI:** 10.3390/ijerph17249532

**Published:** 2020-12-19

**Authors:** Masatsugu Orui, Chihiro Nakayama, Nobuaki Moriyama, Masaharu Tsubokura, Kiyotaka Watanabe, Takeo Nakayama, Minoru Sugita, Seiji Yasumura

**Affiliations:** 1Department of Public Health, Fukushima Medical University School of Medicine, Fukushima 960-1295, Japan; nakac@fmu.ac.jp (C.N.); moriyama@fmu.ac.jp (N.M.); yasumura@fmu.ac.jp (S.Y.); 2Sendai City Mental Health and Welfare Center, Sendai 980-0845, Japan; 3Department of Radiation Health Management, Fukushima Medical University School of Medicine, Fukushima 960-1295, Japan; tsubokura_tky@me.com; 4Department of Medicine, Teikyo University School of Medicine, Tokyo 173-8606, Japan; kiyowata@med.teikyo-u.ac.jp; 5Department of Health Informatics, School of Public Health, Kyoto University, Kyoto 606-8501, Japan; nakayama.takeo.4a@kyoto-u.ac.jp; 6Toho University, Tokyo 143-8540, Japan; sugitamnr@a05.itscom.net

**Keywords:** Fukushima nuclear accident, recovery phase, psychological distress, post-traumatic stress, community mental health services

## Abstract

*Objective:* The Fukushima Daiichi nuclear disaster in 2011 produced psychological reactions among evacuees. Despite the harsh situation, subsequently, there has been gradual progress in reconstruction, with more than half of the evacuees returning after the evacuation. Our hypothesis is that evacuee mental health will now be better due to new stable living conditions. This study aims to clarify the statuses of psychological distress, post-traumatic stress, and radiation health anxiety among evacuees who have rebuilt permanent homes after evacuation. *Methods:* A cross-sectional questionnaire survey of 1600 residents was conducted in 2020. As primary outcomes, the survey measured psychological distress (Kessler 6), post-traumatic stress (post-traumatic stress four-item checklist), and radiation health anxiety. The data are compared for residents who have rebuilt permanent home and those who did not evacuate. *Results:* In the co-variant analysis, the statuses of psychological distress (*p* < 0.001), post-traumatic stress (*p* < 0.001), and radiation health anxiety (*p* < 0.001) are found to still be high, with significant differences when compared to those who did not evacuate. These results are still at an equivalent level for the continuing evacuation. *Conclusion:* Our findings may indicate a necessity for continuing disaster-related mental health activities even though the living conditions have improved.

## 1. Introduction

The Great East Japan Earthquake occurred on 11 March 2011 and generated a massive tsunami that led to enormous damage along the Pacific coast in Japan. This was followed by a separate tsunami which hit the Fukushima Daiichi Nuclear Power Plant (operated by the Tokyo Electric Power Company), leading to a radiation disaster in the Fukushima Prefecture. The Japanese government designated evacuation areas in which residents were forced to relocate to non-evacuation areas and were not allowed to stay overnight after the disaster (Figure 1). Therefore, the long-term evacuation of residents from many surrounding municipalities was required. According to previous reports from officials, there were no direct death cases from low-dose radiation exposure [1]. Regarding morbidity for particular cancers, on-going discussion is required to identify the potential association between exposed and absorbed doses and the risk of developing cancers [2].

The multiple disasters relating to the earthquake, tsunami, and nuclear accident have produced damaging psychological reactions among the evacuees and residents of Fukushima, including post-traumatic stress responses [3,4,5], losses of family, relatives and friends [6], and radiation health anxiety due to low-dose radiation exposure [7,8]. Mental health has also been affected by the loss of employment and/or community ties due to the nuclear disaster and subsequent residential relocation, with consequent drastic changes in living circumstances [9]. It is possible that these situations have led to the increased suicide rates in the evacuation area [10].

Despite this harsh situation, there has been gradual progress in the reconstruction of Fukushima in the past nine years following the nuclear disaster [11,12]. Regarding reconstruction progress, the evacuation orders have been gradually lifted across the region (e.g., Hirono in 2012; Tamura in 2014; Naraha in 2015; Katsurao, Kawauchi, and Minami-Soma in 2016; and Kawamata, Namie, Iitate, and Tomioka and 2017). Only Okuma, Futaba, and partial areas in other municipalities still have orders in place. Therefore, most forced evacuees have been able to return home and rebuild their permanent home (Figure 2). The free provision of temporary housing ended after a grace period of 2–3 years after the lifting of the evacuation orders. Therefore, the majority of forced evacuees in the evacuation area built their permanent home at the point of conducting the survey presented here. Under the present reconstruction conditions, evacuees can choose to return to their original home in the ex-evacuation area or live in a new location outside of the ex-evacuation area. A previous study reported that 80% of evacuees had recovered their prior mental health status, compared to 84.4% for those who had lived in the non-evacuation area [13]. Moreover, returnees who have returned to their original home show a significantly better psychological distress status when compared to evacuees [14]. Hence, our hypothesis is that evacuees who have been able to rebuild their permanent home have better and more stable living conditions than when under evacuation conditions, and the progress of reconstruction leads to an improvement in returnee mental health.

In order to assess the necessity of further disaster-related mental health activities for evacuees and residents, this study aims to clarify the current statuses of psychological distress, post-traumatic stress, and radiation health anxiety among evacuees who have rebuilt their permanent home after the lifting of evacuation orders, comparing their results with those of the residents who did not evacuate. Moreover, evacuees exhibit effects in terms of their sleeping conditions or drinking behavior due to disaster-related experiences or following drastic lifestyle changes [15,16,17,18]. Since health anxiety due to radiation exposure has been associated with media utilization in previous studies [19,20], we also investigate media utilization regarding radiation in order to assess the relationship with radiation health anxiety. Therefore, current lifestyle and media utilization are thought to be useful for understanding evacuee mental health, and these factors are also investigated here. These findings will likely be useful for ongoing disaster risk reduction and management.

## 2. Materials and Methods

### 2.1. Participants

The cross-sectional questionnaire survey targeted 1600 residents of Fukushima Prefecture that were aged between 20 to 80 years. We selected 400 people from the evacuation area. As a sample in the non-evacuation area, 1200 residents from each of the three areas (Hama-Dori, Naka-Dori, and Aizu with 400 people per area) were selected (Figure 1). Participant selection was based on a two-stage stratified random sampling (stage one survey of the region, stage two of individuals). Thirty to thirty-one individuals per point were randomly selected from municipal resident registration files to obtain 1600 representative participants. We sent an anonymous, self-reporting postal questionnaire to subjects in January during 2020. The survey was approved by the ethics review committee of the Fukushima Medical University on 16 July 2019 (approval number 2019-110).

### 2.2. Survey Variables

#### 2.2.1. Current Lifestyle

Current sleep satisfaction was assessed on a four-point scale ranging from “really satisfied”, “satisfied”, “dissatisfied” and “really dissatisfied”. We defined “dissatisfaction with current sleep condition” as those who answered “dissatisfied” or “really dissatisfied”.

Current drinking behavior was assessed with as follows: “Do you drink alcohol every day more than the proper amount (2 drinks, 20 g/day)?” We defined the proper amount for drinking alcohol as less than two drinks based on the “National Health Promotion Movement in the 21st Century (Health Japan 21)” that was produced by the Ministry of Health, Labor, and Welfare [21]. Incidentally, two drinks was defined as 60 mL of spirits (e.g., whiskey or brandy), 240 mL of wine, 500 mL of beer, or 180 mL of Japanese sake.

#### 2.2.2. Psychological Distress

To assess current psychological distress among evacuees and residents, the Kessler 6 (K6) scale was used. The K6 scale is utilized to screen for non-specific serious mental illnesses, including DSM-IV mood and anxiety disorders, which indicate the status of psychological distress within the last 30 days. The score range in the K6 scale is from 0 to 24 points. Those scoring 0–4 points were classified as probably having no psychological distress, and those scoring 5–12 points were classified as having probable mild–moderate stress, while 13 points or more was defined as serious psychological distress [22]. This study used the Japanese version of the K6 scale, which has been empirically validated as an independent means of screening for mental distress among evacuees [23,24].

#### 2.2.3. Post-Traumatic Stress

To measure recent post-traumatic stress among evacuees and residents from the past month, the four-item PTSD (Post-traumatic Stress Disorder) checklist (four-item PCL) was utilized. It was originally based on the PTSD Checklist-Specific (PCL-S) comprising 17 items assessing PTSD symptoms, all of which are rated on a Likert scale from 1 (“not at all”) to 5 (“extremely”) [25]. The Japanese version of the PCL-S has been previously validated [26,27]. In this study, we used the abbreviated version of PCL-S, which is composed of four measurements (re-experiencing: repeated, disturbing, and unwanted memories of the stressful experience; physical reactions: having strong physical reactions when something reminded you of the stressful experience, for example, heart pounding, trouble breathing, sweating; avoidance: avoiding external reminders of the stressful experience; and difficulty concentrating: having difficulty concentrating). The score range for the four-item PCL is from 4 to 20 points and the checklist has proven reliability and validity [28,29].

#### 2.2.4. Health Anxiety Due to Radiation Exposure and Media Utilization about Radiation

For current anxiety levels regarding radiation health risks, participants were asked to subjectively rate “your current level of anxiety about the effects of radiation on your health due to the nuclear disaster” on a five-point scale ranging from “not at all,” “only a little,” “somewhat,” “very,” and “extremely.” “Very” and “extremely” were categorized as the “current strong anxiety group” while other levels of anxiety were categorized as the “no or weak anxiety group” as per a previous study [19]. This questionnaire was investigator-designed.

For the utilization of media about radiation, respondents selected up to three items from the following 13 options: local newspapers, national newspapers, NHK television (public broadcast television, both national and local), private local broadcast television, private national broadcast television, radio, Internet news, Internet sites/blogs, social network services (SNSs), magazines/books, public relations information from local governments, word of mouth, and none of the above. To assess the association between media utilization and current strong health anxiety, we categorized “any local media (local newspaper and broadcasting),” “any national media (national newspaper and broadcasting),” “public broadcasting (NHK),” “any Internet media (Internet news, Internet sites/blogs, SNSs),” and “public relations information from local governments” as dependent variables.

#### 2.2.5. Classification for Forced or Voluntary Evacuation for Past and Ongoing Experiences

We classified forced or voluntary evacuation by combining two factors, i.e., (A) living in an evacuation area as of 11 March 2011 or not and (B) evacuation experience among residents in the non-evacuation area to escape radiation exposure.

The questionnaire data were also divided by evacuation type into five categories. Firstly, we classified forced evacuation (or forced evacuees), which refers to evacuation due to living in an area designated by the national government as an evacuation area as of 11 March 2011. Forced evacuees were divided into two categories, i.e., (1) those who had “rebuilt their permanent home” for those who had already rebuilt their own house or had a rental apartment, and (2) “continuing forced evacuation” for those who were still evacuated, living in public restored housing, or a relative’s house. Moreover, among the residents who had lived in the non-evacuation area, we categorized them into three groups, i.e., (3) “continuing voluntary evacuation”, which refers to voluntary evacuation to avoid the effects of the nuclear disaster, even among residents who lived in non-evacuation areas as of March 2011, and they still have continued evacuation. (4) “Temporary voluntary evacuation” was defined as those who only evacuated voluntarily immediately after the nuclear power plant accident and then had returned home. Those who had no evacuation experience were defined as (5) “no evacuation experience”.

### 2.3. Statistical Analysis

A chi-square test was used to examine the basic characteristics, current lifestyle, and media utilization regarding radiation. Additionally, analysis of covariance (ANCOVA) was used to examine the current statuses of psychological distress, post-traumatic stress, and radiation health anxiety while adjusting for age and gender. For the group between “rebuilt their permanent home” and “no-experience of evacuation”, which featured a particularly large number of respondents, the K6, four-item PCL, and radiation health anxiety scores were compared directly. Regarding subjects who did not provide their age or gender, we excluded these cases. As the purpose of the survey was to compare the mental health statuses under the equivalent conditions between “continuing forced evacuation”, “continuing voluntary evacuation”, and “temporary voluntary evacuation” when not analyzed by ANCOVA and with the “rebuilt their permanent home” and “no evacuation experience” conditions as analyzed by ANCOVA.

Statistical significance was evaluated using two-sided, design-based tests with a 5% level of significance with Stata 15 (StataCorp, 2017. Stata Statistical Software: Release 15. College Station, TX: StataCorp LLC).

## 3. Results

### 3.1. Participants

We sent out 1600 questionnaires from January to March 2020 and received 737 responses, with 145 from the evacuation area and 592 from non-evacuation areas (i.e., a response rate of 46.1%). Then, we excluded 42 respondents who failed to provide information regarding their gender or age, as well as 26 respondents who did not answer the questions about their current living situation among evacuees in the evacuation and evacuation after the nuclear accident among residents in the non-evacuation area. Additionally, respondents were excluded where only their families were evacuated (*n* = 8). The final study population consisted of 661 respondents who had lived in the evacuation area or non-evacuation area (evacuation area, *n* = 137; non-evacuation area, *n* = 524) (Figure 3).

Based on the classification for the forced or voluntary evacuation with past and ongoing experiences, respondents were classified as either having (1) rebuilt their permanent home after evacuation (*n* = 119 residents) or (2) continuing forced evacuation (*n* = 18 residents). Among residents in the non-evacuation area, we considered those (3) continuing voluntary evacuation (*n* = 12) and (4) temporary voluntary evacuation (*n* = 23). Four hundred and eighty-nine residents had no evacuation experience (Figure 3).

### 3.2. Respondent Characteristics

The proportion of respondents aged 65 years and older was higher among evacuees in the evacuation area compared to continuing and temporary voluntary evacuees. Among the voluntary evacuees, more than half of the continuing and temporary voluntary evacuees lived in Hama-Dori or Naka-Dori, which are areas close to the Fukushima Daiichi Nuclear Power Plant. The proportions for unemployed respondents were higher for those who have rebuilt their permanent home or continued forced evacuation (Table 1).

### 3.3. Current Lifestyle and Strong Radiation Health Anxiety and Media Utilization about Radiation

Regarding current sleeping conditions and drinking behavior, there were no significant differences between forced and voluntary evacuees and those with no experience of evacuation.

Moreover, the proportion of those who had perceived radiation health anxiety strongly (i.e., answering “high” or “extremely”) was high among residents who had experienced forced or voluntary evacuation. Regarding media utilization for radiation, although the factor showed a significant association in a previous study [19], there were no significant differences between the five groups (except the utilization of national media) in this study (Table 2).

### 3.4. Psychological Distress Current Lifestyle Health Anxiety Due to Radiation Exposure

Psychological distress (as shown by the K6 score) was higher for those that had rebuilt their permanent home compared to the group with no evacuation experience. The ANCOVA (adjusted by age and gender) showed significant differences between the five categorized groups.

The four-item PCL scores were also higher for those who had rebuilt their own house than those with no evacuation experience. Again, the ANCOVA showed significant differences between the five categories.

Finally, regarding radiation health anxiety, those who had experienced forced or voluntary evacuation (except those in continuing forced evacuation) featured higher radiation health anxiety when compared with those with no evacuation experience (Table 3).

### 3.5. Comparison between Those Who Have Rebuilt Their Permanent Home and Those with No Evacuation Experience

Comparing the values for those who rebuilt their permanent home and those with no evacuation experience under the ANCOVA (adjusted by age and gender), the statuses of psychological distress (K6 score), post-traumatic stress (four-item PCL score), and radiation health anxiety were significantly higher for those who rebuilt their permanent home compared with those with no evacuation experience (psychological distress, *p* < 0.001, F value (1, 587) = 32.0; post-traumatic stress, *p* < 0.001, F value (1, 590) = 48.2; radiation health anxiety, *p* < 0.001, F value (1, 595) = 24.2) (Figure 4).

## 4. Discussion

The present study has aimed to clarify the current statuses of psychological distress, post-traumatic stress, and radiation health anxiety among residents who have rebuilt their permanent home after the lifting of evacuation orders in the Fukushima Prefecture. As a result, those that have rebuilt their permanent home still show a high psychological distress, post-traumatic stress, and radiation health anxiety when compared with those with no evacuation experience, even when they could rebuild their permanent home after the lifting of the evacuation orders.

### 4.1. Previous and Current Statuses of Psychological Distress, Post-Traumatic Stress, and Radiation Health Anxiety among Forced Evacuees

For forced evacuees in the Fukushima Prefecture, Yabe et al. reported that the proportion of adults who scored above the K6 cut-off (≥13) for general mental health was higher than usual, indicating severe mental health problems among evacuees. Moreover, the traumatic symptoms were almost equal to those for workers after the 9/11 World Trade Center attacks [3]. Therefore, it was shown that forced evacuees experienced damage to their mental health following the earthquake, tsunami, and nuclear disaster.

In a previous study, the proportion of strong radiation health anxiety was 19.0% (24/153) in 2016 among forced evacuees [19]. In our findings, the proportion of strong radiation anxiety among residents who had rebuilt their permanent home was 12.9%, which may show a declining proportion of strong radiation health anxiety since 2016 [19].

### 4.2. Lifestyle and Psychological Distress of Evacuees Who Have Rebuilt Their Permanent Home

Among each of the groups, i.e., those who rebuilt their permanent home, those who were still continuing forced or voluntary evacuation, and those had no experience of evacuation, their sleeping and drinking behaviors were not significantly different. In previous studies, evacuees exhibited effects on their sleeping condition or drinking behavior due to disaster-related experiences or following drastic changes in their lifestyle [15,16,17,18]. According to the report from the National Reconstruction Agency in Japan, 70–80% of residents had already built their permanent home (i.e., those living in their own house or a rental apartment) [30], which was almost equivalent to the results from our survey in the evacuation area (18/138 = 86.2%). Therefore, our findings regarding lifestyle improvement might be based on the fact that their living environment has improved after a long period of time since the nuclear power plant accident.

Despite the vast majority of respondents rebuilding their permanent home among evacuees, the current statuses of psychological distress and post-traumatic stress were still high in comparison to those with no evacuation experience, which was an opposite finding to our hypothesis. It goes without saying that the continuing forced/voluntary evacuation groups feature high psychological distress, but it was unexpected that psychological distress has remained high for respondents after rebuilding their permanent home. Murakami et al. reported the prevalence of K6 ≥ 10 (as an indicator of mood/anxiety disorders) among evacuees to be 22.5% at the point of their survey in January in 2018. Also, the prevalence for returnees was 16.2%, which was higher than the value (10.3%) for the whole of Japan [14]. Notably, the prevalence of K6 ≥ 10 among the rebuilt permanent home group in this study was 22.3% (data not shown). The status of psychological distress is still high among those who have rebuilt their permanent home, even after two years have passed since the study by Murakami et al. The reason for this is uncertain; however, in a previous study conducted in the Miyagi Prefecture, which was affected by the tsunami disaster following the 2011 Great East Japan earthquake, the suicide rates in the tsunami disaster-affected area showed a tendency to gradually re-increase during the recovery phase of the disaster when the provision of free temporary housing was terminated. This may have been affected by the consequential increased financial hardship among evacuees. Cutting social ties once again for evacuees after removing their temporary housing may also have been an influence due to relocation to another residence [31,32]. In our findings, even if evacuees had rebuilt their permanent home, it was considered that psychological distress remained at an equivalent level to those who were continuing evacuation as the termination of free temporary housing may have caused loss of social ties or financial hardship. In particular, the proportion of unemployed people among the rebuilt permanent home group was high.

### 4.3. Post-Traumatic Stress and Radiation Health Anxiety of Evacuees Who Have Rebuilt Their Permanent Home

It could be expected that the continuing voluntary evacuation group has continued to have high post-traumatic stress and radiation anxiety because they have evacuated in order to avoid the effects of the nuclear disaster; however, both post-traumatic stress and radiation health anxiety in the rebuilt permanent home group showed significant high scores when compared to the no evacuation experience group. It was a notable finding that both the post-traumatic stress and radiation health anxiety have remained high after evacuees have returned and rebuilt their permanent homes. Those who have rebuilt their permanent home have chosen to return to their original home in the ex-evacuation area after the evacuation orders have been lifted or to live outside of the ex-evacuation area. Therefore, not all forced evacuees have returned to the ex-evacuation area which once featured high levels of radiation. A previous study reported that the environmental radiation levels at the time of the survey were more strongly associated with radiation anxiety than radiation levels immediately after the accident [33]. Moreover, a higher risk perception of radiation exposure after the nuclear accident was associated with later post-traumatic stress symptoms in previous studies [34]. Therefore, the rebuilt permanent home group might show the highest post-traumatic stress among the five groups due to high radiation health anxiety and the risk of a high environmental radiation dose level.

Regarding media utilization about radiation, there were few significant differences among the five groups considered in this study. The reason for this might be that broadcasting about the Fukushima Daiichi Nuclear Power Plant accident has gradually declined when compared to the frequent broadcasting immediately after the accident [35].

### 4.4. Limitations and Strengths

This study has several limitations. First, due to its cross-sectional design, causality could not be established here. Second, the number of respondents for the continuing forced/voluntary evacuation and temporary voluntary evacuation groups were extremely small, therefore, it was impossible to carry out a detailed analysis (e.g., a comparison of five groups); however, the “continuing forced evacuation”, “continuing voluntary evacuation” and “temporary voluntary evacuation” groups are essential subjects when evaluating whether it is necessary to continue support efforts for them in relation to disaster-related mental health. Therefore, we reserve this classification for future research. The third limitation was that our primary outcome, i.e., post-traumatic stress (four-item PCL), has yet to be validated in a Japanese context. Therefore, further studies are needed to confirm the validity and reliability of the Japanese version of the four-item PCL. Fourth, the group of people who have rebuilt their permanent home consisted here of residents who have returned to their original home in the ex-evacuation area (returnees) or those who now live outside of ex-evacuation area. Unfortunately, we could not determine whether the current permanent home was in the ex-evacuation area or not, which was a severe limitation in this study. The fifth limitation was a shortage of queries regarding economic status among subjects in this study (e.g., disaster-related loss of employment or affording to live in current economic status), which may be influential in terms of analyzing psychological distress. Sixth, recall effects should be considered as we used a self-administered questionnaire survey. As the subjects answered for themselves regarding the current situation for most of the queries at the point of the survey, it is considered that the influence of this bias was small. Finally, the response rate was less than 50%, which presents an issue regarding the representativeness of the analyzed group. Previous studies have reported that mental health status might affect the response rate to a survey, suggesting that non-response might be related to mental health status [36]. Many evacuees in with poor mental health may not have been able to answer the survey, and this could have led to underestimations in our analysis.

Despite these limitations, our study can indicate that the prevalence of psychological distress, post-traumatic stress, and radiation health anxiety remains high among evacuees who have rebuilt permanent home, even when their living condition has improved after evacuation. Few studies have reported the current statuses of psychological distress, post-traumatic stress, and radiation health anxiety in the period following the lifting of evacuation orders following the Fukushima Nuclear Power Plant accident [14,37]. With the novel coronavirus (COVID-19) pandemic, it is uncertain whether the pandemic will be influential in terms of the mental health among evacuees and residents. Therefore, it may be necessary to conduct further studies to assess mental health conditions in the area.

## 5. Conclusions

Our findings suggest that the current statuses of psychological distress, post-traumatic stress, and radiation health anxiety remain high for those that have rebuilt their permanent home after the lifting of evacuation orders. This result was opposite to our hypothesis, which proposed that those who had rebuilt permanent home would have reduced psychological distress, post-traumatic stress, and radiation health anxiety. In conclusion, our findings may indicate a necessity for continuing to provide disaster-related mental health services for residents (e.g., counseling or training for counselors) [38,39], even after residents rebuild their permanent home following evacuation.

## Figures and Tables

**Figure 1 ijerph-17-09532-f001:**
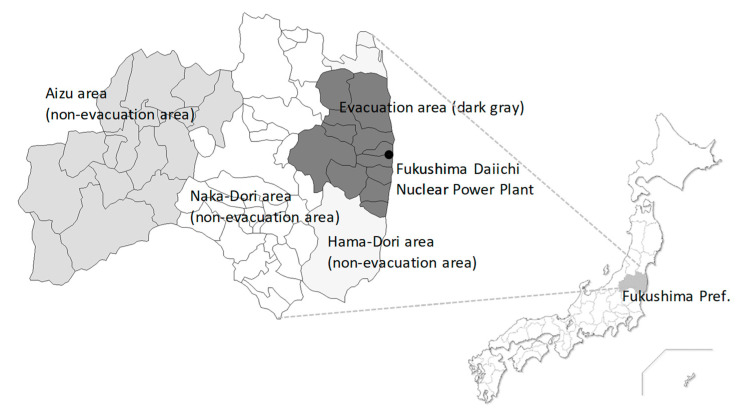
Evacuation and non-evacuation areas around Fukushima. Regions colored in dark gray correspond to municipalities where evacuation orders were issued. Hama-Dori, Naka-Dori, and Aizu were the non-evacuation areas.

**Figure 2 ijerph-17-09532-f002:**
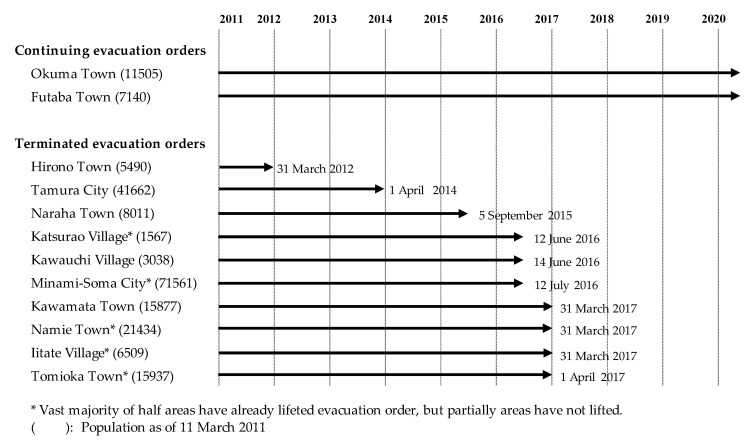
Continuing and lifting evacuation orders in each municipality in the evacuation area.

**Figure 3 ijerph-17-09532-f003:**
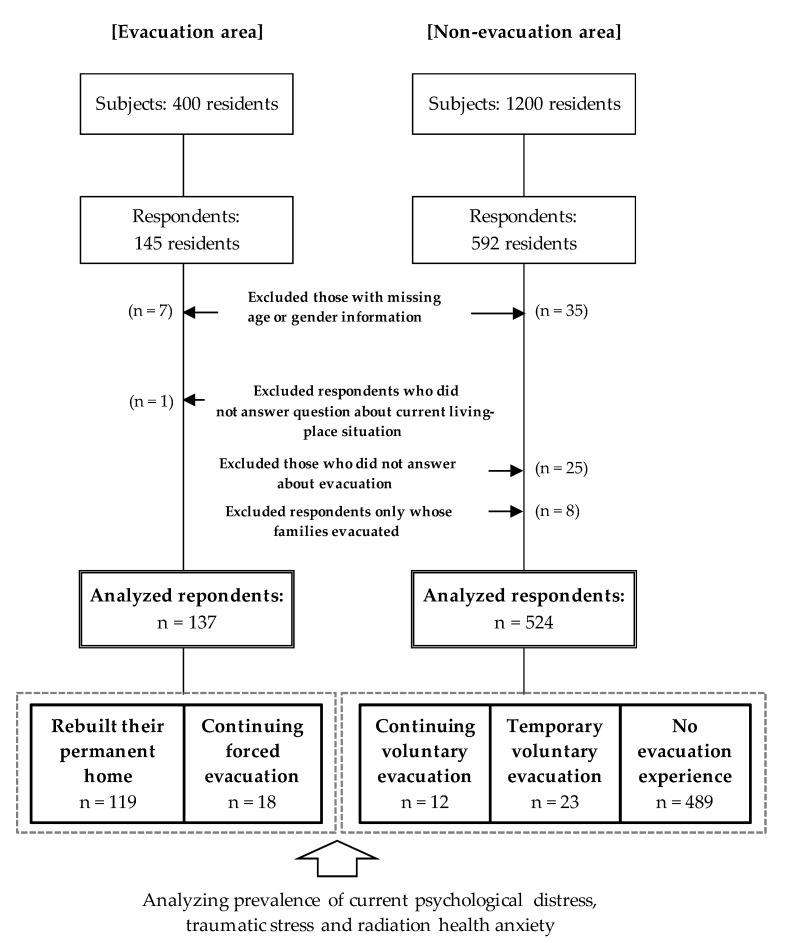
Sample selection for the evacuation and non-evacuation areas and the classification of the current evacuation situation. There are 137 subjects among residents in the evacuation area and 524 residents in the non-evacuation area. These subjects are classified into the five categories shown above.

**Figure 4 ijerph-17-09532-f004:**
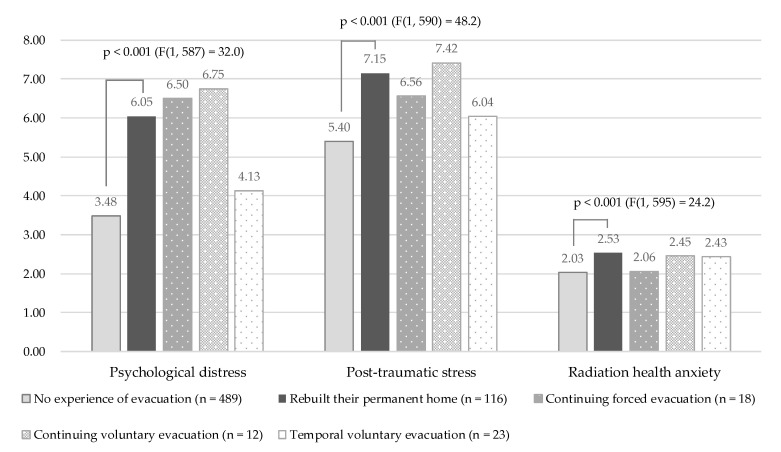
Comparison between those who rebuilt their permanent home and those with no evacuation experience in terms of psychological distress, post-traumatic stress, and radiation health anxiety.

**Table 1 ijerph-17-09532-t001:** Basic characteristics of respondents.

		Evacuation Experience (+)				Evacuation Experience (−) (*n* = 489)
		Forced Evacuation		Voluntary Evacuation		
		Rebuild Permanent Home	Continuing Forced Evacuation	Continuing Voluntary Evacuation	Temporary Voluntary Evacuation	
		(*n* = 119)	(*n* = 18)	(*n* = 20)	(*n* = 23)	
Age	20–39 (%)	13 (10.9%)	0 (0.0%)	4 (20.0%)	3 (13.0%)	72 (14.7%)
	40–59 (%)	25 (21.0%)	4 (22.2%)	5 (25.0%)	11 (47.8%)	148 (30.3%)
	60–80 (%)	81 (68.1%)	14 (77.8%)	11 (55.0%)	9 (39.1%)	269 (55.0%)
Gender	Male (%)	66 (55.5%)	9 (50.0%)	10 (50.0%)	9 (39.1%)	227 (46.4%)
Living area	Evacuation area (%)	119 (100.0%)	18 (100.0%)	0 (0.0%)	0 (0.0%)	0 (0.0%)
	Hama-Dori (%)	0 (0.0%)	0 (0.0%)	9 (45.0%)	16 (69.6%)	139 (28.4%)
	Naka-Dori (%)	0 (0.0%)	0 (0.0%)	10 (50.0%)	7 (30.4%)	174 (35.6%)
	Aizu (%)	0 (0.0%)	0 (0.0%)	1 (5.0%)	0 (0.0%)	176 (36.0%)

**Table 2 ijerph-17-09532-t002:** Current lifestyle, strong radiation health anxiety, and media utilization about radiation.

		Evacuation Experience (+)				Evacuation Experience (−)	*p*-Value (χ^2^ Test)
		Forced Evacuation		Voluntary Evacuation			
		Rebuild Permanent Home	Continuing Forced Evacuation	Continuing Voluntary Evacuation	Temporary Voluntary Evacuation		
		(*n* = 119)	(*n* = 18)	(*n* = 12)	(*n* = 23)	(*n* = 489)	
Current lifestyle							
Sleep condition	Dissatisfied (%)	24 (20.5%)	1 (5.6%)	1 (10.0%)	3 (13.0%)	64 (13.1%)	*p* = 0.233 (χ^2^ = 5.58)
Drinking behavior	Above proper drinking level (%)	42 (35.6%)	6 (33.3%)	2 (20.0%)	5 (21.7%)	135 (27.7%)	*p* = 0.402 (χ^2^ = 4.03)
Health anxiety due to radiation exposure							
Strong radiation health anxiety	Very and extremely high (%)	15 (12.9%)	2 (11.8%)	1 (9.1%)	3 (13.0%)	26 (5.4%)	*p* = 0.044 (χ^2^ = 9.81)
Media utilization about radiation	Local media (%)	81 (68.1%)	15 (83.3%)	7 (58.3%)	16 (70.0%)	384 (78.5%)	*p* = 0.062 (χ^2^ = 8.95)
	National media (%)	28 (23.5%)	9 (50.0%)	4 (33.3%)	8 (35.0%)	214 (43.8%)	*p* = 0.001 (χ^2^ = 17.6)
	Public broadcasting, (NHK) (%)	68 (57.1%)	10 (55.6%)	4 (33.3%)	11 (47.8%)	266 (54.4%)	*p* = 0.572 (χ^2^ = 2.92)
	Any Internet media (%)	28 (23.5%)	3 (16.7%)	5 (41.7%)	10 (43.5%)	151 (30.9%)	*p* = 0.171 (χ^2^ = 6.40)
	Public relations from local government (%)	46 (38.7%)	6 (33.3%)	4 (33.3%)	7 (30.4%)	138 (28.3%)	*p* = 0.229 (χ^2^ = 5.63)

**Table 3 ijerph-17-09532-t003:** Psychological distress, post-traumatic stress, and radiation health anxiety by lifestyle.

		Evacuation Experience (+)				Evacuation Experience (−)	*p*-Value (ANCOVA) *
		Forced Evacuation		Voluntary Evacuation			
		Rebuilt Permanent Home	Continuing Forced Evacuation	Continuing Voluntary Evacuation	Temporary Voluntary Evacuation		
		(*n* = 119)	(*n* =18)	(*n* = 12)	(*n* = 23)	(*n* = 489)	
Psychological distress (K6)	Mean (SD)	6.05 (5.50)	6.50 (5.90)	6.75 (5.55)	4.13 (3.70)	3.48 (4.11)	*p* < 0.001 (F(4, 64) = 10.29)
Post-traumatic stress (four-item PCL)	Mean (SD)	7.15 (3.26)	6.56 (2.36)	7.42 (3.75)	6.04 (3.84)	5.40 (2.08)	*p* < 0.001 (F(4, 64) = 13.02)
Radiation health anxiety	Mean (SD)	2.53 (1.07)	2.06 (1.20)	2.45 (1.04)	2.43 (1.12)	2.03 (1.00)	*p* < 0.001 (F(4, 64) = 6.67)

SD: Standard deviation; * ANCOVA: Analysis of covariance (adjusted by gender and age).

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
