# Peer review of "Current Psychological Distress, Post-traumatic Stress, and Radiation Health Anxiety Remain High for Those Who Have Rebuilt Permanent Homes Following the Fukushima Nuclear Disaster"

_ijerph, 2020, doi:10.3390/ijerph17249532_

Round 1

Reviewer 1 Report

Relevance is very high from the point of view the possible radiation emergencies in future. Sample is unique. Designe and methodological approach are adequate. Statistial analysis is OK. Results and their interpretation are grounded.

Author Response

Reviewer 1

Comments and Suggestions for Authors

  • Relevance is very high from the point of view the possible radiation emergencies in future. Sample is unique. Designe and methodological approach are adequate. Statistial analysis is OK. Results and their interpretation are grounded.
    • We really appreciate for your reviewing.

Reviewer 2 Report

Dear Authors,

Thank you very much for an interesting and important study. Attached, please find several suggestions that would improve the manuscript:

  1. Since you measured Health anxiety, pleas add to the introduction a description of morbidity and mortality among the exposed population (evacuated vs. non-evacuated)
  2. Besides, all content you measured has to be explained in the introduction. The introduction should include a description of lifestyles, alcohol use, and link with media utilization. For example, the sentences: 139-141: "Since health anxiety due to radiation exposure was associated with media utilization in previous studies [22,23], we also investigated media utilization about radiation in order to assess the relation to radiation health anxiety" should be located in the introduction, rather than the Methods.

Methods

  1. The first section in the Methods (with the two figures) should be written in the introduction.
  2. I don't understand why you exclude participants with no age or gender from the database- Especially that you didn't measure these variables.

Result

In my opinion, the division you made into four groups does not benefit from the manuscript. Since the groups are so small, the results cannot be inferred (I agree that mental health status might affect the response rate, as you mentioned in the discussion). Maybe you could divide it into the main two groups and considered a multivariate analysis to understand if participants' location is a significant variable. Besides, if you have empty cells in the Chi-square test, there is no meaning for these results.

Author Response

Reviewer 2

Comments and Suggestions for Authors

  • Thank you very much for an interesting and important study. Attached, please find several suggestions that would improve the manuscript:

  1. Since you measured Health anxiety, pleas add to the introduction a description of morbidity and mortality among the exposed population (evacuated vs. non-evacuated)
  • As you pointed out, we have added the text as followed:

According to previous reports from officials, there were no direct death cases from low-dose radiation exposure [1]. Regarding morbidity for particular cancers, on-going discussion is required to identify of the potential association between exposed and absorbed doses and the risk of developing cancers [2].

  1. Besides, all content you measured has to be explained in the introduction. The introduction should include a description of lifestyles, alcohol use, and link with media utilization. For example, the sentences: 139-141: "Since health anxiety due to radiation exposure was associated with media utilization in previous studies [22,23], we also investigated media utilization about radiation in order to assess the relation to radiation health anxiety" should be located in the introduction, rather than the Methods.
  • As you pointed out, we have modified the introduction section. (L83)

(Methods)

  1. The first section in the Methods (with the two figures) should be written in the introduction.
  • As you pointed out, we have modified the introduction section. (L61)

  1. I don't understand why you exclude participants with no age or gender from the database- Especially that you didn't measure these variables.
  • As you pointed out, it is necessary to explain why we excluded participants with no age or gender.

Therefore, we have added the text in the Methods section. (L174)

Regarding subjects who did not provide their age or gender, we excluded these cases. As the purpose of the survey was to compare the mental health statuses under the equivalent conditions between “continuing forced evacuation”, “continuing voluntary evacuation”, and “temporary voluntary evacuation” when not analyzed by ANCOVA and with the “rebuilt their permanent home” and “no evacuation experience” conditions as analyzed by ANCOVA.

(Result)

  1. In my opinion, the division you made into four groups does not benefit from the manuscript. Since the groups are so small, the results cannot be inferred (I agree that mental health status might affect the response rate, as you mentioned in the discussion). Maybe you could divide it into the main two groups and considered a multivariate analysis to understand if participants' location is a significant variable. Besides, if you have empty cells in the Chi-square test, there is no meaning for these results.
  • As you pointed out, dividing into four groups may not a benefit because of small number. However, "continuing forced evacuation", "continuing voluntary evacuation" and "temporary voluntary evacuation" must be essential subjects if we evaluate whether it is necessary to continue supports for them as disaster mental health activities or not. Therefore, we would like to remain this classification for future research reference.

Reviewer 3 Report

This cross-sectional study, entitled “Current psychological distress, traumatic stress, and radiation health anxiety remain high those who have rebuilt permanent home following Fukushima

nuclear disaster”, assesses the correlates of mental health among the Fukushima nuclear disaster evacuees who have rebuilt their permanent homes 9 years after the disaster (year 2020). Nuclear disasters rarely occur, and this study will provide important evidence in thinking about the association between long-term mental health outcomes and household status of the evacuees. Overall, the study is carefully designed and analyzed. I have several comments that might help improve this paper.

Major comments

  1. Throughout the manuscript (including the manuscript title), I noticed so many grammatical/style flaws (too many to mention). I strongly urge the authors to use an English language editing service.
  2. Abstract: It would be helpful for the readers to know the numbers (N and other data) in the abstract.

Minor comments

  1. Throughout the manuscript, the authors are inconsistent in reporting year/month/dates (e.g., “March 11, 2011”, “2017.4.1”, “11 March, 2011”).
  2. Throughout the manuscript, the authors use a term “traumatic stress”. Given the relationship between the disaster and the subject responses, I think this phrase should be reworded to “posttraumatic stress”.
  3. Introduction, page 2, lines 46-47: To my understanding, not all residents were exposed to excessive radiation. May I suggest the authors to change “radiation exposure” to “potential radiation exposure”?
  4. Materials and Methods, page 3, line 91: I was unable to understand the recruitment process. In particular, could you clarify the following sentence: “We selected 400 people from the evacuation area”? Did you select a total 1200 people from each of the three evacuation areas (Hama-Dori, Naka-Dori, and Aizu: 400 people per area)?
  5. Discussion: Is it worthwhile to clarify that this survey was conducted in prior to the COVID-19 pandemic? The participants were likely to report mental health results differently after the pandemic.

Author Response

Reviewer 3

Comments and Suggestions for Authors

  • This cross-sectional study, entitled “Current psychological distress, traumatic stress, and radiation health anxiety remain high those who have rebuilt permanent home following Fukushima nuclear disaster”, assesses the correlates of mental health among the Fukushima nuclear disaster evacuees who have rebuilt their permanent homes 9 years after the disaster (year 2020). Nuclear disasters rarely occur, and this study will provide important evidence in thinking about the association between long-term mental health outcomes and household status of the evacuees. Overall, the study is carefully designed and analyzed. I have several comments that might help improve this paper.

(Major comments)

  1. Throughout the manuscript (including the manuscript title), I noticed so many grammatical/style flaws (too many to mention). I strongly urge the authors to use an English language editing service.
    • Thank you for your comments. As you pointed out, we have requested "English edit service" to MDPI and have made proofread our manuscript throughout.

  1. Abstract: It would be helpful for the readers to know the numbers (N and other data) in the abstract.
    • As you pointed out, we have added the subjects number and analysis results.

(Minor comments)

  1. Throughout the manuscript, the authors are inconsistent in reporting year/month/dates (e.g., “March 11, 2011”, “2017.4.1”, “11 March, 2011”).
    • Thank you for your comments. We have revised the order to “Month” “Day”, “Year” consistently.

  1. Throughout the manuscript, the authors use a term “traumatic stress”. Given the relationship between the disaster and the subject responses, I think this phrase should be reworded to “posttraumatic stress”.
    • As you pointed out, we have changed "post-traumatic stress" from "traumatic stress" throughout.

  1. Introduction, page 2, lines 46-47: To my understanding, not all residents were exposed to excessive radiation. May I suggest the authors to change “radiation exposure” to “potential radiation exposure”?
    • Thank you for your comments. As you pointed out, not all residents were exposed to excessive radiation, but they exposed low-dose radiation. Therefore, we have changed "low-dose radiation exposure" from "radiation exposure". (L56)

  1. Materials and Methods, page 3, line 91: I was unable to understand the recruitment process. In particular, could you clarify the following sentence: “We selected 400 people from the evacuation area”? Did you select a total 1200 people from each of the three evacuation areas (Hama-Dori, Naka-Dori, and Aizu: 400 people per area)?
    • As you pointed out, we selected a total 1200 residents from each of the three area (Hama-Dori, Naka-Dori, and Aizu: 400 people per area). Therefore, we have revised it as followed:

As a sample in the non-evacuation area, total 1200 residents from each of the three area (Hama-Dori, Naka-Dori, and Aizu: 400 people per area) were selected (Figure 1). (L93)

  1. Discussion: Is it worthwhile to clarify that this survey was conducted in prior to the COVID-19 pandemic? The participants were likely to report mental health results differently after the pandemic.Thank you very much for an interesting and important study. Attached, please find several suggestions that would improve the manuscript:
  • As this study was conducted in prior to the COVID-19 pandemic, it is uncertain whether it will be influent to mental health status among evacuees and residents. Therefore, it may be necessary to conduct further studies to assess these condition. (L349)

With the novel coronavirus (COVID-19) pandemic, it is uncertain whether the pandemic will be influential in terms of the mental health among evacuees and residents. Therefore, it may be necessary to conduct further studies to assess mental health conditions in the area.